# Experiences of a National Early Warning Score (NEWS) intervention in care homes during the COVID-19 pandemic: a qualitative interview study

Rachel Stocker ,[1] Siân Russell,[1] Jennifer Liddle ,[1,2] Robert O Barker,[1] Adam Remmer,[3] Joanne Gray,[4] Barbara Hanratty ,[1,2] Joy Adamson[5]

[1]Population Health Sciences Institute, Newcastle University, Newcastle upon Tyne, UK
[2]Applied Research Collaboration North East and North Cumbria, Newcastle upon Tyne, UK
[3]Community Services, Specialist Older Person Team, South Tyneside and Sunderland NHS Foundation Trust, Tyne and Wear, UK
[4]Department of Nursing, Midwifery & Health, Northumbria University, Newcastle upon Tyne, UK
[5]Department of Health Sciences, University of York, York, UK

**Correspondence to**
Dr Rachel Stocker;
rachel.stocker@newcastle.ac.uk

## ABSTRACT

**Background** The COVID-19 pandemic has taken a heavy toll on the care home sector, with residents accounting for up to half of all deaths in Europe. The response to acute illness in care homes plays a particularly important role in the care of residents during a pandemic. Digital recording of a National Early Warning Score (NEWS), which involves the measurement of physical observations, started in care homes in one area of England in 2016. Implementation of a NEWS intervention (including equipment, training and support) was accelerated early in the pandemic, despite limited evidence for its use in the care home setting.

**Objectives** To understand how a NEWS intervention has been used in care homes in one area of North-East England during the COVID-19 pandemic, and how it has influenced resident care, from the perspective of stakeholders involved in care delivery and commissioning.

**Methods** A qualitative interview study with care home (n=10) and National Health Service (n=7) staff. Data were analysed using thematic analysis.

**Results** Use of the NEWS intervention in care homes in this area accelerated during the COVID-19 pandemic. Stakeholders felt that NEWS, and its associated education and support package, improved the response of care homes and healthcare professionals to deterioration in residents' health during the pandemic. Healthcare professionals valued the ability to remotely monitor resident observations, which facilitated triage and treatment decisions. Care home staff felt empowered by NEWS, providing a common clinical language to communicate concerns with external services, acting as an adjunct to staff intuition of resident deterioration.

**Conclusions** The NEWS intervention formed an important part of the care home response to COVID-19 in the study area. Positive staff perceptions now need to be supplemented with data on the impact on resident health and well-being, workload, and service utilisation, during the pandemic and beyond.

## INTRODUCTION

The care home sector is one of the most overlooked components of health and social care in the UK. Care homes provide accommodation, personal care and support, and companionship to some of the most vulnerable, frail and medically complex older adults in society. There are more than 11 000 care homes (nursing and non-nursing) in the UK,[1] home to over 400 000 residents, representing 16% of those aged 85 or over. The sector is dominated by private, for-profit, providers, who provide around four-fifths of care home beds in the UK.[2] Care homes may receive some funding from the NHS, however, the majority of funding comes from local authorities, or from residents or residents' families. Care home residents receive medical care from visiting National Health Service (NHS) primary care professionals, including general practitioners and community nurses. Care configuration of NHS services for care home residents varies significantly across the country. Some areas provide a varied multidisciplinary healthcare team to support care home residents; others focus on specialist nurse provision; others have no special service.[3]

Care homes have been severely affected by the COVID-19 pandemic. The virus spread

### Strengths and limitations of this study

► This study is the first to examine the role of a National Early Warning Score (NEWS) intervention in care homes during the COVID-19 pandemic in the UK.

► Care home and National Health Service staff were recruited and interviewed in May 2020, providing an important insight into the COVID-19 response in this sector.

► The role of NEWS in identifying COVID-19 disease and monitoring resident health, was explored in-depth with semi-structured remotely conducted interviews.

► General practitioners and care home residents were not recruited for practical reasons; future research should directly seek rather than infer their views.

rapidly and many residents died in a short space of time. Estimates suggest that up to half of all COVID-19 related deaths in Europe were care home residents,[4] and people working in social care are dying at two times the rate of the general population.[5] This situation has been described by the WHO as an 'unimaginable human tragedy'.[6] Many care homes have struggled with staffing, accessing personal protective equipment[7] and maintaining isolation measures during the pandemic.

In recent years, efforts have been made to support the care home sector to improve the quality of health-related care.[8] Identifying and monitoring residents for early signs of acute illness, using tools such as the National Early Warning Score (NEWS; the latest version of the scoring system is the 'NEWS2'), has been a particular area of interest.[9] The NEWS was initially developed for use in hospital settings. It requires the measurement of six parameters: temperature, pulse, systolic blood pressure, respiratory rate, oxygen saturation and level of consciousness. The overall NEWS triggers a response, ranging from repeating the measurements within a specific time frame, to initiating an emergency medical response. The NEWS is intended to serve as an adjunct to, as opposed to replacing, a clinical decision.[9]

Implementation of the NEWS into care homes is already underway across multiple regions in the UK,[10 11] despite a lack of validation and proven effectiveness outside the hospital setting.[12] The aim is to improve the response to acute illness in care home residents by improving triage and communication with external healthcare professionals, and to reduce avoidable hospital admissions. In care homes, the NEWS intervention has generally been implemented as a 'package', which includes as a minimum the training and equipment to measure the vital observations needed to generate the NEWS itself. Some implementations of a NEWS package have additionally included an ongoing support programme to care homes, and the use of a cloud-based storage system to store (and sometimes transmit to primary care staff for use in triage) NEWSs. Concerns have been expressed about the appropriateness of using the NEWS in care homes, including potential adverse influences on palliative care.[13] NEWS was also designed for use by clinically trained staff, such as registered nurses, who are not on-site in residential care facilities. Previous work indicates that while the collection of NEWS in care homes appears to be feasible,[14] the complexity of the care home context and subsequent intricacies in implementing NEWS need careful thought and planning to exploit any potential benefits for residents.[15 16] Early in the COVID-19 pandemic in the UK, the British Geriatrics Society emphasised the importance of care home staff having the skills, training, and equipment to identify deterioration in residents, and advocated the use of systems that incorporate NEWS.[17]

This study focuses on the ways in which one Clinical Commissioning Group (CCG) with a high level of socio-economic deprivation in the North-East of England, used their NEWS intervention in response to the COVID-19

---

**Box 1   NEWS intervention**

**Pre-COVID-19**
► Equipment for taking vital signs and digital tablet for measuring and recording NEWS.
► Education and support delivered by an NHS clinical educator with a nursing background covering: practical use of equipment, interpretation of individual vital signs and resulting NEWSs, contextualising NEWS observations and underpinning scoring system with the importance of 'soft signs' of deterioration.
► Internet web-based cloud storage system, whereby NEWSs are automatically uploaded from the tablet to the storage system. The NHS clinical educator and other NHS staff involved in care for each care home can access the web-based system to view NEWSs.

**Additional function in response to COVID-19**
► Additional functionality offered by the equipment was unlocked thereby expanding the NEWS package in this area further—notably a picture-taking facility to upload pictures of wounds.

NEWS, National Early Warning Score; NHS, National Health Service.

---

pandemic. Having initially implemented a NEWS intervention in 2016, providing care homes with the equipment for taking vital signs, a digital tablet for recording NEWS and associated training, in 2019 the CCG invested further when a nurse was appointed as a clinical educator to provide more tailored training and support to care home staff (see box 1 for an outline of the NEWS intervention in this area). While most care homes in the CCG had received the 2019 intervention, for those who had not the CCG accelerated the roll out of this, with the aim of ensuring all residential and nursing homes had the equipment and training necessary to measure residents' physical observations and calculate a NEWS.

This paper presents a qualitative interview study with key stakeholders, with the aim of understanding how a NEWS intervention has been used in care homes during the COVID-19 pandemic in the UK, and how it has influenced resident care. Specific research questions were:

1. How have care home staff used the NEWS intervention to assess resident illness during the COVID-19 pandemic?
2. What is the experience of care home staff of using the NEWS intervention during the COVID-19 pandemic?
3. How has the COVID-19 pandemic influenced the roll out of the NEWS intervention in this area?

## METHODS

Semi-structured interviews were conducted with stakeholders, including care home staff, NHS healthcare professionals and commissioners, in a CCG area in the North-East of England. This area is one of the 20% most deprived in England with high rates of morbidity and a lower-than-average life expectancy, with a resulting pressure on care home bed availability. Stakeholders were identified using purposive sampling, through existing links with local NHS staff. Participants were also asked to identify other colleagues involved in the

## Box 2 Interview topics

- ► Perceived validity of NEWS in care homes.
- ► Experiences of using NEWS before, and during the pandemic.
- ► NEWS training for care homes.
- ► Perceptions on the usefulness of NEWS during the pandemic, including identification of potential cases.
- ► How NEWS influenced decision making and triage, within and outwith care homes.
- ► Variations in use of NEWS during the pandemic.

NEWS, National Early Warning Score.

NEWS intervention in this area (snowball sampling). We sought to include staff from a range of residential and nursing care homes, particularly with varied experience using NEWS, including long-standing users and users who received the intervention as part of the accelerated roll-out in response to COVID-19. A key NHS employee, known to potential participants, acted as a gatekeeper, sending introductory emails with a participant information sheet. Once consent to be contacted by the research team was secured, a researcher (RS), made direct contact by email to invite participation. All individuals approached agreed to take part. Informed consent was secured either electronically or verbally in line with Health Research Authority principles.

All interviews were carried out remotely by telephone or videoconferencing, during May 2020. Development of an interview topic guide (box 2) was informed by previous research conducted by the team, published literature evaluating the use of NEWS in care homes, and relevant literature and direct communications to the research team describing how care home staff were experiencing the pandemic in the UK. Field notes were taken after interviews.

Interviews were carried out by RS or SR (both female research associates, doctorates in health services research and medical sociology respectively, and experience of qualitative studies in care homes), audio-recorded and transcribed verbatim. All transcripts were anonymised. Interviews lasted 30–60 min.

### Patient and public involvement

Patients and the public were first involved in this study at the idea generation stage, via the Newcastle University Care Home Interest PPI Group (CHIG). The research questions, topic guide and study design were discussed via email with group members before the study commenced. CHIG members stressed the importance of carrying out interviews remotely and flexibly to time constraints, given the burden of the pandemic on the health and social care sector. Our discussions with this PPI group confirmed that COVID-19 related research in care homes is viewed as a pressing research priority. PPI partners will be consulted on appropriate dissemination strategies.

### Data analysis

RS and SR conducted a thematic analysis of the dataset, following the principles of Braun and Clark's six-phase framework.[18] An inductive, data-driven approach was taken to analysis, meaning that the themes generated through our analysis were identified from the data itself, rather than via the use of a pre-existing coding frame.[19] We chose this approach to collect rich data on our topic, using a contextualist approach which 'acknowledges the ways individuals make meaning of their experience, and, in turn, the ways the broader social context impinges on those meanings, while retaining focus on the material and other limits of 'reality' (p9).[18] NVivo V.11 software was used to aid in data management. Data analysis started once the first interview was carried out, and was carried out iteratively, throughout the interviewing period and beyond.

Transcripts were read in full (phase 1—familiarisation with data), then coded line by line, separately, by RS and SR (phase 2—generating initial codes). Codes were compared within and between transcripts, using the constant comparative technique,[20] to streamline the coding framework and identify themes (phase 3—search for themes). Emergent themes were discussed within the study team (phase 4—reviewing themes) and linked together to form a final set of themes and subthemes (phase 5—defining themes) which were then situated within each of the three key research questions for the purposes of reporting (phase 6—writing up analysis). Interviews ceased when no new semantic codes, describing the experiences of staff, were identified from the data (code saturation[19]) as agreed by RS and SR.

## RESULTS

We interviewed 17 stakeholders, comprising 10 from the care home sector across 7 care homes, and 7 from the NHS (see table 1). Of the seven care homes, five provided both residential and nursing care and two provided solely residential care (see table 2 for care home characteristics). There are no care homes in this area which provide nursing care only. One care home received the accelerated form of the NEWS training and implementation during the COVID-19 pandemic in the UK, the remainder had had NEWS implemented pre-pandemic.

Higher level themes are organised according to how they address each of the key research questions, which are presented in turn.

### How have care home staff used the NEWS intervention to assess resident illness during the COVID-19 pandemic?

NEWS intervention during the pandemic: an adjunct to COVID-19 identification

Many stakeholders described how, prior to the COVID-19 outbreak, the use of the NEWS package had become embedded and habitualised into the everyday routines of care homes, and staff interactions with external healthcare services. Care homes recorded NEWS baseline

**Table 1** Interviewees and their working role

| Working role | Sector | n | Description |
|---|---|---|---|
| Care home manager *Nursing background* *Non-nursing background* | Care home | 7 (3) (4) | Managers oversee the running of the home ensuring that guidelines and regulations are adhered to. |
| Care home senior carer/nurse | Care home | 3 | Senior carers provide care and limited healthcare to residents and oversee the care provided by junior carers. Care home nurses provide healthcare to residents. |
| Director | NHS | 1 | Director within a regional NHS Foundation Trust. Provided advice to the implementation team on the use of NEWS within the context of patient safety and reducing avoidable hospital admissions. |
| Director | NHS-related body | 1 | Supported the implementation team and aided networking and learning between various CCGs and academic institutions. |
| NHS nurses *Specialist older persons' nurse* *Clinical educator* | NHS | 4 (3) (1) | Specialist nurses visit care homes to provide healthcare to care home residents. They provide a link between the care homes and external services, aiming to prevent avoidable hospital admissions. The clinical educator provided training to the care homes as well as ongoing support and monitoring. |
| Commissioning of NHS services for older people | NHS | 1 | Programme manager with a focus on technological innovation in care settings leading the implementation of NEWS in care homes. |
| | Total | 17 | |

NEWS, National Early Warning Score; NHS, National Health Service.

**Table 2** Care home characteristics

| Care home | Care provided | Number of beds |
|---|---|---|
| Care home 1 | General care and specialist dementia | ~50 |
| Care home 2 | General care and specialist dementia | ~50 |
| Care home 3 | General care and specialist dementia | ~30 |
| Care home 4 | Predominately specialist dementia | ~40 |
| Care home 5 | Specialist dementia and end of life care | ~40 |
| Care home 6 | Specialist dementia | ~40 |
| Care home 7 | General care and nursing | ~30 |

scores regularly (for routine monitoring and comparison when residents became acutely unwell), except for those near the end of life. A NEWS was frequently requested by community healthcare professionals responding to concerns from care homes, and staff often calculated a NEWS before contacting healthcare professionals.

At the outbreak of the COVID-19 pandemic in the UK and the start of lockdown measures, use of the NEWS by care homes in this area increased substantially. Using NEWS became increasingly embedded into daily care home and NHS practices, and was viewed as a key aspect of the pandemic response in this setting. Care home staff, NHS healthcare professionals and service commissioners shared a belief that measuring NEWS, its component physiological observations, alongside the associated education and support package, have enhanced the response to deteriorating resident health during the pandemic.

Taking vital signs observations and using NEWS with the specific intention of identifying possible COVID-19 infection varied across the care homes in our study. Stakeholders highlighted the power of the NEWS in identifying early illness due to COVID-19 as well as other causes, and the ability of the NEWS intervention to facilitate remote monitoring, thereby decreasing the need for healthcare professionals to visit care homes.

*Use of individual physiological observations to identify COVID-19*
If COVID-19 was suspected, care home staff frequently measured temperature and oxygen saturations to use as individual physiological measures, as opposed to the NEWS. The use of these physiological observations for identifying possible COVID-19 disease was advocated by a local NHS network. Care home staff appreciated the importance of responding to isolated abnormalities in temperature or oxygen saturations to help identify possible COVID-19, even if the NEWS was not triggering a high level of concern. Stakeholders remained broadly sceptical about the sensitivity and specificity of these clinical observations to accurately help the identification of COVID-19 in this population—feeling that they went some way to help, and could prompt further investigation of potential COVID-19, but could not be conclusive.

> [There are] two main symptoms with our experience, high temperature and low oxygen saturations. We've found in a couple of our residents, […] they had no temperature, and no breathing difficulty, but I couldn't understand—and when we checked their NEWS score we found oxygen saturation was low. So it helps us in the early diagnosis and to do appropriate support for the residents. (Care home manager 1)

> The response to the infection is hugely physiological. So NEWS is a way of prompting you to do physiology. […] I think if you'd asked me six months ago I'd have said 'Actually the NEWS score is more useful than the underlying physiology'. Actually, I'd be tempted to go the other way around now and say 'You do a NEWS

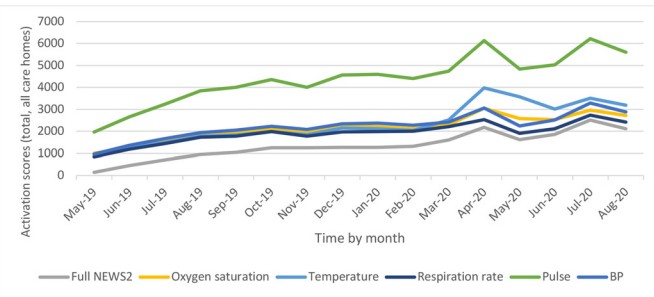

**Figure 1** Frequency of NEWS2 and its component measures taken in care homes in the study area. BP, blood pressure; NEWS2, latest version of the scoring system 'National Early Warning Score'.

score because it makes you look at the physiology and you're going to go, oh my god, look at that oxygen saturation'. (NHS director)

Frequency of NEWS recordings and individual observations significantly increased month-on-month from March 2020 onwards, with some care homes taking baseline NEWSs several times a week. Other care homes recorded the NEWS sporadically, suggesting wide variation in the use of NEWS in this area during the UK pandemic. Figure 1 illustrates the increase in NEWS and individual observations recording before and during the pandemic.

### Using the NEWS intervention to remotely monitor health

It was felt among care home and NHS healthcare professionals that, prior to the pandemic, NEWS could speed up initial triage and facilitate remote treatment decisions, expediting the process for residents to receive the care they required. Referrals accompanied by a NEWS reduced the need for clinicians to visit care homes to take physical observations. Identification of early illness was considered to be more efficient with the use of NEWS. Set within this context, it is perhaps not surprising that another positive aspect of the NEWS intervention during the pandemic was to facilitate remote monitoring of residents' health by healthcare professionals. This was believed to have reduced unnecessary footfall and thus infection risk, which was a major driver to the accelerated roll out of the NEWS intervention during the pandemic.

We had to just take a stand and reflect and say, 'What do we need to do? We know that the predictions [are that COVID-19] is going to come. We're likely to have to go into lockdown. Nobody is going to be able to go into care homes'. So we thought, 'actually NEWS is an important tool in that. […] We just need to respond'. (NHS commissioner)

CCGs have been formally told to set up remote MDT care home rounds, so they are having to put the IT in place. […] You need a digital way of recording [resident] observations, and then you can put in a safety netting process that will allow the care home to get back in touch. Because there's going to be loads

of [residents] where you're going to go, 'they're not quite right but I definitely don't want to send them to hospital'. (NHS director)

The ability for NHS staff, such as community nurses, to remotely access physiological observations and compare with baselines, was of particular value. The camera facility of the NEWS tablet was also unlocked in response to the pandemic, giving external healthcare professionals the ability to assess and treat wounds remotely.

[int: Are there any doctors or GPs who are taking it upon themselves to log on to the [cloud system] and look at the NEWS scores?] Yes, absolutely. [Pre-pandemic] I don't think it was utilised [by them], especially not as much as it could have been, but due to the COVID crisis, everyone across the city now want access to it and it's obviously taken a pandemic to realise how valuable the tool is. (NHS specialist nurse 1)

I think the biggest change for the [use of the NEWS intervention] at the moment is the pictures. […] For example, if there was a patient who had a skin flap […] We'll say, 'Will you take a photo? Will you upload it so we can see it?' We can physically see that photo and then we can assess the severity, we can assess when the visit should next take place […] it's brilliant for that aspect. (NHS specialist nurse 2)

### What is the experience of care home staff of using the NEWS intervention during the COVID-19 pandemic?

#### Empowerment of care home staff with the NHS agenda and language

Using the NEWS intervention to help identify and triage responses to deteriorating resident health was described by stakeholders as empowering. For care home staff, the NEWS intervention represented a helpful link to the NHS. The NEWS intervention represented objective clinical information which care home staff could generally rely on to back up their intuition that the resident was unwell (termed 'soft signs' for example, decreased appetite). Having a common clinical language which care home staff could use—or leverage for action, where necessary—was useful to promote integrated working between care homes and the NHS. It also facilitated triage to appropriate services, often negating the need for a primary care healthcare professional to attend in person, which is an important consideration during the pandemic.

Care home staff felt confident that with a NEWS reading, their concerns about deteriorating residents would be taken seriously by external healthcare professionals, as it evidenced their concerns. This was compared positively to care home staff's previous experiences seeking help from primary care, where staff often felt that concerns were not listened to. This feeling of empowerment was capitalised on by both care homes and local NHS services during the COVID-19 pandemic.

When we've rang the likes of home recovery or 111 and we're challenged as to why we're ringing, when we can't really put a finger on it that somebody's not well. Now, this [NEWS] has really aided us to be able to look for those signs, look for those symptoms, using the NEWS score and being able to give better communication across the telephone as to why and what has happened with that particular resident. I know there's been times where we've kind of been brushed off by services. And I'm a big believer in our care staff, they may not be qualified nurses but they know their residents sometimes better than we do. [...] Now I feel like we've got this tool now to be able to kind of prove that somewhat as well, and it does help back up what we're saying when it comes to our residents. So it's been great. (Care home manager 2)

During the COVID-19 pandemic, care home staff were relied on to take on clinical procedures and observations usually carried out by visiting NHS professionals. This challenge to traditional role boundaries was welcomed and encouraged by NHS stakeholders, as it met the needs of the NHS to reduce footfall into care homes and limit the number of hospital admissions. In the early days of the pandemic, and without straightforward guidance for care homes, care home staff also welcomed this extension to their roles. They were able to measure and focus on clinical observations—in effect, the NEWS intervention facilitated a more objective and decisive response within care homes to tackling a then-new virus.

[int: What do you think about the use of NEWS during this pandemic in care homes?] Well I've always loved it. I really love the NEWS tablet and for why, because I think it totally empowers the care staff. [...] Care staff, when they say, 'They're just not right. I don't know what's wrong with them, but the person's not right'. Now they've got a little more ammunition to say, 'well actually, such-and-such isn't very well. These are the observations …' and it makes them have more of a voice. That's what I love about it. So when they ring the GP and they say, 'Well actually the observations are blah, blah, blah', they're almost listened to a little bit more because they've got some sort of clinical information rather than just saying, 'Oh I don't know what's wrong with them'. I think that applies to here and now as well with the pandemic. (NHS specialist nurse 2)

### How has the COVID-19 pandemic influenced the roll out of the NEWS intervention?

#### Centrality of training relationships and clinical support for accelerated implementation of NEWS during a pandemic

When the pandemic broke out in the UK, the CCG commenced a rapid roll out of the NEWS intervention to the minority of care homes in the area that were waiting for implementation. Care homes already using the NEWS intervention before the pandemic were contacted by the clinical educator to reinforce that support was available. The pandemic highlighted the importance of care homes, particularly those without registered nursing staff, having an engaged and supportive nominated NHS clinical educator.

#### Training and implementing the NEWS intervention in care homes during the pandemic

Training to support rapid implementation during the pandemic consisted of a short, socially distanced visit from the clinical educator to the care home, training one or two key members of staff in the home rather than a larger body of staff, and without a practical component. This rapid roll out model demonstrated that relationships built during training were integral to ongoing NEWS use and understanding. The training constraints related to the pandemic were perceived as having had a major impact on the quality of the rapid implementation and confidence of staff.

[During the rapid training], I don't feel like I built great rapport. I don't feel like it instilled that much confidence in them and I haven't really received many phone calls from those homes. Yes they're doing NEWS, but usually the homes that I've really been visibly in, really supported them and built those rapports and stuff up with, I feel like I get more phone calls which is great because they want support which then that in turn means they've picked up something, they're looking for answers and help and they get it. Where those other homes, I would worry that maybe they don't feel as confident to pick up the phone and speak with me because I've raced in, did what I had to do and then left—and it's very difficult to build up rapport over time, and trust. (NHS specialist nurse 1)

#### Ongoing clinical support for care homes during the pandemic

Emphasis was placed on the importance of ongoing clinical support for care homes. In this area, care home staff had access to the clinical educator and a specialist nurse with a specific care home remit. Having these key contacts, who were knowledgeable and approachable, was valued by care home staff and capitalised on during the COVID-19 pandemic. They were able to build on long-term relationships with each other, cultivating understanding of each sector and respect for each other's role and remit.

This is not just about the technical or the medical aspect, it's often about the relationships that [clinical educator] is then building up. [...] I think that methodology has worked because you're consolidating it with a really good grip and really getting [care homes] to be the expert. (NHS commissioner)

[Clinical educator] is a very approachable person. You can email him, you phone him and in all fairness, we would probably see [clinical educator] at least probably once a week. He'll pop in just to make

sure everything is alright and we're well, you know, and if we're having any problems. So that support is continually there. (Care home manager 4)

This cross-sector team working helped to legitimise use of NEWS in a non-NHS setting, and smoothed many processes of integrated care over traditional NHS and social care boundaries.

## DISCUSSION
### Summary of findings
The roll out of the NEWS intervention in the study area was accelerated by COVID-19. Care home staff and NHS commissioners viewed it as an important part of a combined care home-NHS response to care delivery during the pandemic. The NEWS intervention was used as an adjunct to carer identification of deteriorating resident health, whether due to COVID-19 or other acute illnesses. It was perceived to have facilitated remote decision-making by healthcare and care home staff and minimised footfall in care homes during the pandemic. In relation to possible COVID-19 infection, care home staff reported increased reliance on specific physiological measures such as oxygen saturations, as opposed to the overall NEWS. Care home staff felt empowered by their extended role in the measurement of physiological observations; better able to respond to resident deterioration and communicate their concerns to healthcare professionals. The intervention encompassed training and support from a clinical educator and sharing of clinical measurements and NEWS with health services. Dis-entangling the relative contributions of education, technology and clinical assessment to resident care is challenging.

### Comparison with other work
Previous evaluations of the use of NEWS in a non-hospital setting reflect many of our findings, including the benefits of using a common clinical language, and facilitating communication between sectors.[21] The importance of using NEWS as a part of a wider assessment of health, rather than a ritualistic, task-oriented procedure,[22] has been stressed by both the UK Royal College of Physicians[9] and those evaluating its use in other settings including primary[13] and community care.[21] The clinician-led training and ongoing support model for the NEWS intervention in place in this area addressed these concerns.

There is some evidence to support the use of NEWS in the identification of and triage of hospital patients with suspected COVID-19.[23 24] However, there have been calls for NEWS to be modified with a more sensitive score for oxygen demand to better account for the development of hypoxic respiratory failure in COVID-19 patients.[25] Evidence for the use of NEWS in community settings during the pandemic remains sparse. There is little comparable evidence available to evaluate the utility of NEWS in care homes specifically during the COVID-19 pandemic.

This intervention led to an increase in assessment and monitoring of a vulnerable population of care home residents during a global pandemic. Greater vigilance and attention to physiological measures within care homes may be the critical component, irrespective of the detail of NEWS and adherence to an escalation plan. However, this intervention has a number of features that minimised face to face contacts with residents, which should reduce infection risk for residents and staff. This is important, as a high proportion of care home residents are asymptomatic when they test positive for COVID-19,[26] a finding reflected in the experiences of our interviewees. Bluetooth transfer of data from measuring equipment to digital tablet, cloud data storage and the possibility of rapid information transfer to outside agencies, were all of potential benefit during COVID-19. The lack of any requirement to manually input recordings from thermometers and blood pressure monitors was a time saving feature, that promoted accuracy and uptake,[27] and in the pandemic, minimised time in close contact with the resident. It is also noteworthy that during COVID-19, staff often completed a subset of measurements, rather than a complete assessment. Selection of what were perceived to be the most important measures (oxygen saturation and temperature) was a modification of NEWS that further reduced staff resident contact. It is also supported by recent evidence[28] that highlights the potential utility of individual oxygen saturation, respiratory rate and temperature measurements. However, reducing the range of measurements does mean that a NEWS cannot be calculated. Whether the absence of a NEWS per se over and above the individual measurements has any impact on resident care, is an important question for the future.

From an implementation perspective, the training and support for the NEWS intervention was delivered by an experienced NHS nurse who understood the needs of the sector. In terms of work needed to implement change,[27] the organisational burden of planning and implementation for the CCG and NHS clinical educator was minimised as they had already been actively engaged in a revised implementation prior to the pandemic and thus there was a pre-existing awareness and embeddedness of NEWS across the care home sector in the region. Most care home staff in our sample were already familiar with NEWS prior to the pandemic, hence, a shift in roles and identities to take on NEWS related work had already taken place, and only minimal work was needed to engage care home staff in implementation[27] or to change working practices, after the outbreak. This pre-existing familiarity, coupled with the reported ease of use of the NEWS equipment and the support from the wider healthcare system, enhanced the adoption, spread and scale-up[27] of the NEWS intervention in these care homes during the pandemic.

Regular, informal contact from a trusted NHS professional to care homes may have been particularly important

early in the pandemic, when information was sparse and there were many uncertainties about how best to provide care.[29] The focus in training on identifying 'soft signs' of deterioration was also significant. So called 'soft signs'—a resident having a reduced appetite for a favourite food, for example—are not perceived as measurable or objective. However, they are observed by care home staff, and have their own unique importance.[30] Inclusion within the training could be seen as a validation of care home staff contribution to resident care.

Implementation of change in the care home sector is challenging. Resources are limited and pressures on staff in particular, may limit the potential to engage in new initiatives.[31] Engaging and supporting care homes, with a good understanding of their needs, is critical when implementing change.[16 32] In this case, the intervention was introduced from outside the sector, raising the possibility that it would be perceived as irrelevant or inappropriate for use in care homes. Prepandemic and ongoing NHS support for the NEWS intervention legitimised and enhanced its use, reflecting other studies in the care home sector.[33] Widespread changes to working practices during COVID-19 are also likely to have had a major influence on staff readiness to embrace change.

### Strengths and limitations

We were able to interview 10 care home staff from 7 care homes in the locality, and 7 NHS staff, despite the pandemic-related pressure on the NHS and social care sectors during the interviewing period (May 2020). This represents a strength of our study and highlights the importance that interviewees placed on the topic and expressing their views. However, the views of care home staff working in less senior roles, such as non-senior carers, was less represented due to care home staffing pressures. Our data may not reflect all care home experiences in other areas of the country, particularly those who are not working with NEWS. Similar interventions have been introduced across England, but this model is specific to the study area, which makes comparison with cross-sector interventions in other geographical areas more difficult.

Despite our efforts, we were unable to recruit GPs, which we believe was due to the pressure the pandemic had placed on primary care. The pandemic visiting restrictions in care homes also meant that we were unable to interview any care home residents themselves to explore their experiences during the pandemic and their understanding of care had been influenced by the introduction of the NEWS intervention. Future work should aim to include these groups, so that their views are directly included rather than inferred.

### CONCLUSION

The NEWS intervention may have a useful role in care homes during the COVID-19 pandemic, enhancing remote working and offering staff some clinical reassurance and structure to their role. However, it is important to acknowledge the paucity of data on NEWS in care homes, despite increasing uptake. Positive staff perceptions from this study now need to be supplemented with data on the impact on resident health and well-being, workload and service utilisation, during the COVID-19 pandemic and beyond. A multidisciplinary consensus on best practice for NEWS use in this sector is required.

**Contributors** RS, SR and JA designed the study with JL and BH. RS and SR conducted interviews. RS, SR and JA analysed data. RS, SR, JA and BH drafted the article, and AR, JL, JG and ROB performed critical revision of the article for important intellectual content. RS is guarantor of the paper. All authors approved the final version to be published.

**Funding** This study is funded by the National Institute for Health Research (NIHR), Research for Patient Benefit [award number PB-PG-0418-20034]. JL and BH are funded by the NIHR Applied Research Collaboration North East and North Cumbria (ARC NENC).

**Disclaimer** The views expressed are those of the author(s) and not necessarily those of the NIHR or the Department of Health and Social Care.

**Competing interests** None declared.

**Patient and public involvement** Patients and/or the public were involved in the design, or conduct, or reporting, or dissemination plans of this research. Refer to the Methods section for further details.

**Patient consent for publication** Not required.

**Ethics approval** Approval was granted by Newcastle University Faculty of Medical Sciences Research Ethics Committee (ref: 2569/2020) and NHS North of England Commissioning Support; HRA approvals were not needed as the work was deemed to be a service evaluation.

**Provenance and peer review** Not commissioned; externally peer reviewed.

**Data availability statement** No data are available. Data are not publicly available.

**ORCID iDs**
Rachel Stocker http://orcid.org/0000-0002-8189-2746
Jennifer Liddle http://orcid.org/0000-0003-1059-1230
Barbara Hanratty http://orcid.org/0000-0002-3122-7190

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
