## [Reviewer comments · BMJ Open]

ARTICLE DETAILS

TITLE (PROVISIONAL)	Experiences of a National Early Warning Score (NEWS) intervention in care homes during the COVID-19 pandemic: A qualitative interview study
AUTHORS	Stocker, Rachel; Russell, Sian; Liddle, Jennifer; Barker, Robert; Remmer, Adam; Gray, Joanne; Hanratty, Barbara; Adamson, Joy

VERSION 1 – REVIEW

REVIEWER	Sabi Redwood Bristol, Population Health Sciences
REVIEW RETURNED	03-Nov-2020

GENERAL COMMENTS	1. Thank you for asking me to review this very timely paper on the use of the NEWS 2 in care homes, and its accelerated uptake in the wake of the pandemic. The findings reported are interesting. However, the authors need to reflect the small number of participating care homes in one clinical commissioning group in the way that they report the findings and point to their tentative nature. For example, in the first paragraph of the first theme it is difficult to decipher what the authors report as study findings. The change in tense from past to present is also quite confusing. It would be helpful if the provisional nature of the findings were acknowledged in the way findings are reported. 2. A more detailed description of the context of the CCG and the participating care homes would be helpful in understanding the findings. 3. The authors state that the study was underpinned by a phenomenological approach. There is no reference to this approach, nor is there any indication in the subsequent parts of the paper that the authors used any phenomenological methods. There are only 2 references to data analysis and no description of the overall approach the authors used to generate the findings. It would be helpful to get some clarity on the methods used. 4. The authors conclude with the following statement: "It is important to separate out the role of the training, support and equipment, to the NEWS2 itself, when evaluating the efficacy of the intervention. ... The perceived success of the NEWS2 intervention in this CCG may point to the usefulness of the NEWS2 vital observation and cloud storage package itself during the pandemic, rather than the NEWS2 theory itself." This is utterly perplexing and appears to be introducing an entirely new set of assumptions by the authors about 'NEWS2 theory', although it is not clear what they mean by that, as well as the various components of the implementation. The link between the reported findings and this conclusion needs to be clearly articulated. 5. It would help readers of this paper if the authors reduced the number of sentences in which they use the passive voice.
--

REVIEWER	LK Bragstad Universitetet i Oslo Det medisinske fakultet
REVIEW RETURNED	31-Dec-2020

GENERAL COMMENTS	Thanks for the opportunity to review this manuscript. I believe it is important to assess the utility of NEWS and other EWS systems in home care and care home settings. This study may contribute an important insight into this. Overall, I think this manuscript lacks a bit of clarity – in particular in terms of the objectives. As an international reader I would have liked to get a bit more context of the health care service organization of the UK – in particular on the collaborative structures between the care homes and the NHS. Review - bmjopen-2020-045469 Remote assessment of care home residents during the COVID-19 pandemic – what is the role of the National Early Warning Score? A qualitative interview study Reviewer comments Thanks for the opportunity to review this manuscript. I believe it is important to assess the utility of NEWS and other EWS systems in home care and care home settings. This study may contribute an important insight into this. Overall, I think this manuscript lacks a bit of clarity – in particular in terms of the objectives. As an international reader I would have liked to get a bit more context of the health care service organization of the UK – in particular on the collaborative structures between the care homes and the NHS. Further comments follow addressing the reviewer checklist. Review checklist Comments to the items where I have ticked “no”. 1. Is the research question or study objective clearly defined? This is one of my main concerns with this manuscript. There appears to be several different research questions and study objectives stated throughout the manuscript. Looking at the title of the manuscript, I expect the aim and objective to be connected to remote assessment of care home residents using NEWS2. The objective stated in the abstract does not correspond entirely with neither the title nor the aim of the study stated on page 5, line 18-20. This needs to be clarified, and uniform throughout the manuscript. 2. Is the abstract accurate, balanced and complete? Based on the unclear objective of the study and the presentation of aim, results and discussion, the abstract is not sufficiently accurate in my reading. This may need to be revised in line with the revision or clarification of the objectives.
---

	6. Are the outcomes clearly defined? Again, because I feel the aim/objective to be unclear, it is difficult to see how the outcomes are sufficiently clear. Reading the results and the discussion, it becomes unclear to me what the actual outcomes of this study are – there are several outcomes introduced, but they are not sufficiently clear. 9. Do the results address the research question or objective? 10. Are they presented clearly? To me, the results section is not sufficiently clear. In my reading, parts of the results section is concerned with describing the setting and context of the study. At the same time, the background lacks description of the UK setting/context of care homes and how this is organized with regards to the NHS and i.e. nursing resources. For an international reader, this context is important to be able to understand the results. A definition of the different roles of the stakeholders included in the sample may also be appropriate for an international reader – what is the difference between a NHS commissioner as opposed to a NHS director or NHS specialist nurse? However, when this information about the context is intertwined with the results it is difficult to follow clearly what is reporting of results and what is contextual comments. In my reading of this manuscript, it was difficult to understand the four higher level themes – and how these inform the same RQs. This could possibly be more clearly described to make it more easily accessible for the reader. When it comes to the outcomes of the study – this is also partly obscured, as I don't see the logic of the themes coming together to inform the RQs. To me, it appears that the results section address a number of different RQs that are not necessarily stated in the methods section or background. It lacks clarity. 11. Are the discussion and conclusions justified by the results? I find the discussion a bit disconnected from the results section. For instance, the authors discuss the NEWS2 package as a complex intervention (page 14) and they describe how it is “difficult to unpack the black box of improvement”. As I understand this, the authors introduce a new concept to the manuscript which is not thus far described. Evaluating complex interventions would call for a different study design than the one applied here in my opinion. This argument is thus taken out of context in my opinion. Especially when this is position in a section summarizing the findings. I find the discussion, overall, to be a bit underdeveloped. I find it difficult to identify an actual discussion of the present findings. The authors dedicate half of the discussion section to summarizing the findings, rather than discussing them more in depth. What does this mean? How can the findings from this study help advance the practice of identifying acute illness in care homes? How can we understand this as an improvement of current practice during a pandemic? 12. Are the study limitations discussed adequately?
--	--

	To me, there are apparent limitations of this study that is not addressed and discussed. The focus is on the strength of interviewing ten care home staff – however, the stakeholders in the NHS are not mentioned, even though I would expect i.e. NHS nurses and GPs to be important stakeholders here in terms of receiving more accurate patient information. It is unclear how the care home residents would contribute to understanding the role of NEWS2 in remote assessments during the pandemic? I think the limitations would need to be discussed further within the research team, in conjunction with a discussion of the aims and objectives of this study. The limitations should be addressed in terms of how they may have limited the study in reaching its objective.
--	--

VERSION 1 – AUTHOR RESPONSE

Reviewer 1. Dr. Sabi Redwood, Bristol.

Comment raised [ ]	Response by author [ ]
1. Thank you for asking me to review this very timely paper on the use of the NEWS 2 in care homes, and its accelerated uptake in the wake of the pandemic. The findings reported are interesting. However, the authors need to reflect the small number of participating care homes in one clinical commissioning group in the way that they report the findings and point to their tentative nature. For example, in the first paragraph of the first theme it is difficult to decipher what the authors report as study findings. The change in tense from past to present is also quite confusing. It would be helpful if the provisional nature of the findings were acknowledged in the way findings are reported.	Thank you for your helpful review comments on our paper. We have re-written the paper, making major changes to the majority of sections, to address all reviewer comments. We have added an acknowledgement that our findings may not be representative of all care homes in the country in the Strengths and Limitations section. We have also significantly restructured the Findings and Discussion sections to better separate and clarify our findings versus our paper discussion. Significantly, we have removed the first theme, and moved narrative about the study and NEWS context to the Introduction and Methods. We have also amended the tense accordingly. We hope that these significant revisions better separate the context and findings.
2. A more detailed description of the context of the CCG and the participating care homes would be helpful in understanding the findings.	We agree. We have added further detail about the history of the NEWS package in this area in the Introduction section (as above), further contextual information about the socioeconomic status of the CCG area, and a more detailed description of the participating care homes and individual interviewees.
3. The authors state that the study was underpinned by a phenomenological approach. There is no reference to this approach, nor is there any indication in the subsequent parts of the paper that the authors used any phenomenological	Thank you. On reflection we have removed the reference to a phenomenological approach to provide clarity. We carried out a thematic analysis underpinned by Braun and Clarke's approach, detailed in their seminal 2006 paper

methods. There are only 2 references to data analysis and no description of the overall approach the authors used to generate the findings. It would be helpful to get some clarity on the methods used.	(referenced in the manuscript). This is a standard, and well used approach to the analysis of qualitative data. We have described the method further. We have also provided an additional reference on the use of thematic analysis from Braun and Clarke (2019).
4. The authors conclude with the following statement: “It is important to separate out the role of the training, support and equipment, to the NEWS2 itself, when evaluating the efficacy of the intervention. ... The perceived success of the NEWS2 intervention in this CCG may point to the usefulness of the NEWS2 vital observation and cloud storage package itself during the pandemic, rather than the NEWS2 theory itself.” This is utterly perplexing and appears to be introducing an entirely new set of assumptions by the authors about ‘NEWS2 theory’, although it is not clear what they mean by that, as well as the various components of the implementation. The link between the reported findings and this conclusion needs to be clearly articulated.	We agree that we needed to be clearer about what we are referring to in terms of the NEWS theory (we used this term to refer to the NEWS2 scoring system – the NEWS acronym has now been used throughout for consistency and clarity as per the editor’s request), versus the NEWS intervention which refers to the entirety of the implementation package, including the training, equipment, underpinning scoring system, and support package provided by the NHS-based clinical educator. We have also revised our description to the ‘NEWS intervention’ throughout to enhance clarity We have significantly revised the manuscript in several sections to make this distinction much clearer, particularly in the introduction (to set the scene) and associated conclusion. We have also amended the title to further clarify.
5. It would help readers of this paper if the authors reduced the number of sentences in which they use the passive voice.	We have undertaken a re-write of this manuscript and reduced our use of the passive voice.

Reviewer 2. Dr. LK Bragstad , Universitetet i Oslo Det medisinske fakultet.

Thanks for the opportunity to review this manuscript. I believe it is important to assess the utility of NEWS and other EWS systems in home care and care home settings. This study may contribute an important insight into this.

Overall, I think this manuscript lacks a bit of clarity – in particular in terms of the objectives. As an international reader I would have liked to get a bit more context of the health care service organization of the UK – in particular on the collaborative structures between the care homes and the NHS. Comments to the items where I have ticked “no”.

Comment raised	Response by author
1. Is the research question or study objective clearly defined? This is one of my main concerns with this manuscript. There appears to be several different research questions and study objectives stated throughout the manuscript. Looking at the title of the manuscript, I expect the aim and objective to be connected to remote assessment of care home residents using NEWS2.	Thank you for your review comments which we have found very helpful. We have significantly re-written the paper, making major changes to the majority of sections, to address all reviewer comments. We agree that the objective of the study and associated research questions were unclear in the manuscript. We have amended the title, and more clearly

The objective stated in the abstract does not correspond entirely with neither the title nor the aim of the study stated on page 5, line 18-20. This needs to be clarified, and uniform throughout the manuscript.	stated our primary aim and research questions at the end of the Introduction section and in the abstract objective. Our aim was to understand how a National Early Warning Score (NEWS) intervention package has been used in care homes during the COVID-19 pandemic, and how it has influenced resident care. We have also re-structured and revised our themes to directly link back to our research questions.
2. Is the abstract accurate, balanced and complete? Based on the unclear objective of the study and the presentation of aim, results and discussion, the abstract is not sufficiently accurate in my reading. This may need to be revised in line with the revision or clarification of the objectives.	We have revised the abstract as per our response to point 1 to further clarify our objective and research questions.
6. Are the outcomes clearly defined? Again, because I feel the aim/objective to be unclear, it is difficult to see how the outcomes are sufficiently clear. Reading the results and the discussion, it becomes unclear to me what the actual outcomes of this study are – there are several outcomes introduced, but they are not sufficiently clear.	We hope that our clarification of our objective as per the points above, plus the revisions to our results section, provides further clarity.
9. Do the results address the research question or objective? 10. Are they presented clearly? To me, the results section is not sufficiently clear. In my reading, parts of the results section is concerned with describing the setting and context of the study. At the same time, the background lacks description of the UK setting/context of care homes and how this is organized with regards to the NHS and i.e. nursing resources. For an international reader, this context is important to be able to understand the results. A definition of the different roles of the stakeholders included in the sample may also be appropriate for an international reader – what is the difference between a NHS commissioner as opposed to a NHS director or NHS specialist nurse? However, when this information about the context is intertwined with the results it is difficult to follow clearly what is reporting of results and what is contextual comments. In my reading of this manuscript, it was difficult to understand the four higher level themes – and how these inform the same RQs. This could possibly be more clearly described to make it more easily accessible for the reader. When it comes to the outcomes of the study – this is also partly obscured, as I don't see the	We agree that the results and discussion sections required revision. As you identified, the start of the discussion contained text more appropriate for the results section. We have engaged in a significant re-write of our results and discussion sections. Notably, the removal of Theme 1, with important contextual narrative moved into the Introduction/Methods where appropriate. We have also re-written and moved up text from the discussion into the results where appropriate. This re-writing, we believe, has enhanced clarity and further demarcates our results versus our discussion. We have also aligned the discussion with our original research questions and objective, to make the link clearer for the reader. We have also added contextual information about the UK context of care homes and the NHS, and further description of the care home setting in this particular area of England. Further definitions of stakeholder roles have been added to Table 1 and an additional table (Table 2) to describe our care home sample. We hope that the changes to the manuscript to provide more clarity on our objective and associated research questions, in addition to the revisions made to the results and discussion to demarcate each section better, better describe the logic of how our higher-level themes are

logic of the themes coming together to inform the RQs. To me, it appears that the results section address a number of different RQs that are not necessarily stated in the methods section or background. It lacks clarity.	related to our research questions.
11. Are the discussion and conclusions justified by the results? I find the discussion a bit disconnected from the results section. For instance, the authors discuss the NEWS2 package as a complex intervention (page 14) and they describe how it is “difficult to unpack the black box of improvement”. As I understand this, the authors introduce a new concept to the manuscript which is not thus far described. Evaluating complex interventions would call for a different study design than the one applied here in my opinion. This argument is thus taken out of context in my opinion. Especially when this is position in a section summarizing the findings. I find the discussion, overall, to be a bit underdeveloped. I find it difficult to identify an actual discussion of the present findings. The authors dedicate half of the discussion section to summarizing the findings, rather than discussing them more in depth. What does this mean? How can the findings from this study help advance the practice of identifying acute illness in care homes? How can we understand this as an improvement of current practice during a pandemic?	We have significantly revised the discussion section to enhance clarity and address comments pertaining to the split between the results and discussion sections (as above). We have also deleted the reference to the ‘black box of improvement’ and ‘complex interventions’ - we agree that this was not the focus of the manuscript. We have significantly developed and re-written the discussion in response to your helpful pointers and comments.
12. Are the study limitations discussed adequately? To me, there are apparent limitations of this study that is not addressed and discussed. The focus is on the strength of interviewing ten care home staff – however, the stakeholders in the NHS are not mentioned, even though I would expect i.e. NHS nurses and GPs to be important stakeholders here in terms of receiving more accurate patient information. It is unclear how the care home residents would contribute to understanding the role of NEWS2 in remote assessments during the pandemic? I think the limitations would need to be discussed further within the research team, in conjunction with a discussion of the aims and objectives of this study. The limitations should be addressed in terms of how they may have limited the study in reaching its objective.	We have added further discussion about the study limitations in the Discussion, clarifying and highlighting our study sample. As above, we have also engaged in significant re-writing with clarification of our aim and objectives, which we believe has strengthened the manuscript.

VERSION 2 – REVIEW

REVIEWER	Sabi Redwood Bristol, Population Health Sciences
REVIEW RETURNED	25-Apr-2021

GENERAL COMMENTS	Thank you for giving me the opportunity to review this version of the manuscript which I am delighted to say reads much better following the authors' revisions. I suggest the following minor revisions: Page 2 line 50 consider removing 'at the height of the pandemic', keep to May 2020 unless you want to be more specific about the UK situation at that time. Page 6 line 20 importance of care home staff having the skills, training etc - please check manuscript for other instances where reference is made to care homes rather than staff working there. Page 9 line 15 please define and reference data saturation page 11 line 43 onwards this paragraph needs clarification. The term 'identifying COVID19' is confusing because COVID19 infection can be suspected, but it is confirmed through testing. Physiological measures such as temperature and oxygen saturation may be indicative of COVID19. I suggest the authors revise this section and check for medical accuracy. Page 13 line 7 'the NEWS intervention represented 'hard clinical data' which care home staff could generally rely upon'. This sentence needs clarification. Hard clinical data is placed in inverted commas. Does that mean it is a term used by a participant? Are the authors referring to the score providing objective clinical information? Page 12 line 47 care home staff were relied upon? Or were they required to do so to limit contact with NHS staff? Page 12 line 52 the authors refer to the 'early days of the pandemic' and to the pandemic in other parts of the manuscript. I suggest this needs to be set into the UK context as internationally waves of increases in infection rates in hospital admissions varied. Page 12 line 57 I do not understand the sentence; the virus was new, not unknown, and I am not clear what 'facilitated a more concrete response within care homes' means. Page 18 line 41 what does the 'shifting roles and identities' mean? Page 18 line 52 contact from a trusted NHS professional? Page 19 line 12 why would care home staff see an intervention that was introduced from outside their own sector as irrelevant or inappropriate? Page 19 line 31 I suggest revising the sentence about study not being representative. The study was conducted in a particular context and it is this that needs to be considered when looking at issues around transferability. A qualitative study would not be 'representative'. Page 19 line 40 could you clarify whether you set out to include GPs in your study. Did you try to recruit, but had no response? Page 19 line 55 what does 'offering reassurance and structure to their role' mean? – please clarify.
---

REVIEWER	LK Bragstad Universitetet i Oslo Det medisinske fakultet
REVIEW RETURNED	25-Mar-2021

GENERAL COMMENTS	Thanks for the opportunity to review the revised manuscript. I appreciate the authors' responses and changes made in response to the reviewer comments supplied from peer reviewers.
--

The manuscript has gained clarity, and is now much improved in my opinion. I appreciate the efforts made to describe the UK system for an international reader and particularly appreciate table 1 – this adds significant clarity for my reading. Well done on the revision.

I have only minor comments at this point, see comments below.

Methods – Page 9, lines 13-15: The authors state that interviews were ceased when saturation was reached. Personally, I think saturation is a difficult concept, as I am not of the belief that additional interviews would not yield new data. However, my question is whether this was in fact the case – that you conducted transcriptions, and analysis while simultaneously conducting interviews (in May 2020). If that is the case, I think this would be an important piece of information to provide in the methods description. It would add to the transparency of the methods used.

Results – At the end of the results section, as a reader I would appreciate a sentence or two to summarize rather than ending with two quotes. Maybe the first section of the discussion (summary of findings) should indeed be moved to the end of the results section?

Discussion/limitations

Another thing I am curious about is whether the authors have given any thought to how the rapid roll-out of these tools and the guidance and supervision given to home care staff might also impact on the validity of the measurements. As you identify in our paper, these tools were originally developed to be used by a different group of professionals, with other competencies than the staff who are now using these. Any thoughts on this might be useful to share either in limitations or in discussion.

The lack for information from the care staff who used NEWS might also need to be highlighted as a possible limitation, even though you state that the perspectives you sought were from professionals at “higher” levels. My concern is that the administrative and leadership staff may have quite different experiences with these kinds of tools, this has been apparent in other studies evaluating similar tools. My general advice would be to highlight this as a limitation, rather than focus on the lack of information from the care home residents. Would you expect the care home residents of these homes to be able to discuss and give insight into your stated research questions?

I notice that you have a statement about the inability to identify the active ingredients of the intervention. I find this statement a bit peculiar as long as this was not a goal stated in the purpose/aim. Furthermore, I would expect this to be a goal more connected to a different kind of evaluation, using other methods. Your informants wouldn't, in my opinion, be the best source of information of this kind.

Page 18, lines 25-29: You refer to the inability to calculate the NEWS score when the measurements are not complete. I would have liked to see a bit more of your reflections around this. There are studies that indicate that the score itself is not the only important part these tools play. Using the tools to identify change in health status may also contribute valuable information for health care professionals using NEWS and similar EWS tools.

	I still think the limitations of the study could be stated a bit more clearly, based on the objectives of the study.
--	--

VERSION 2 – AUTHOR RESPONSE

Reviewer: 1

Dr. Sabi Redwood, Bristol

Comments to the Author:

Thank you for giving me the opportunity to review this version of the manuscript which I am delighted to say reads much better following the authors' revisions. I suggest the following minor revisions:

Comment raised	Response by author
Page 2 line 50 consider removing 'at the height of the pandemic', keep to May 2020 unless you want to be more specific about the UK situation at that time.	We agree - removed.
Page 6 line 20 importance of care home staff having the skills, training etc - please check manuscript for other instances where reference is made to care homes rather than staff working there.	This has been checked and corrections made as appropriate.
Page 9 line 15 please define and reference data saturation	We have clarified the type of saturation we refer to and have provided a reference. Also please see reply to comment from reviewer 2.
page 11 line 43 onwards this paragraph needs clarification. The term 'identifying COVID19' is confusing because COVID19 infection can be suspected, but it is confirmed through testing. Physiological measures such as temperature and oxygen saturation may be indicative of COVID19. I suggest the authors revise this section and check for medical accuracy.	We have added 'possible COVID-19 disease' to this section, and 'to help identify' - to clarify that physiological measures can point to possible disease (rather than being diagnostic).
Page 13 line 7 'the NEWS intervention represented 'hard clinical data' which care home staff could generally rely upon'. This sentence needs clarification. Hard clinical data is placed in inverted commas. Does that mean it is a term used by a participant? Are the authors referring to the score providing objective clinical information?	Thank you, we have replaced this term with 'objective clinical information' which reads better.
Page 12 line 47 care home staff were relied upon? Or were they required to do so to limit contact with NHS staff?	Added 'upon'
Page 12 line 52 the authors refer to the 'early days of the pandemic' and to the pandemic in other parts of the manuscript. I suggest this needs to be set into the UK context as internationally waves of increases in infection rates in hospital admissions varied.	Added reference to the UK context where appropriate throughout the manuscript.
Page 12 line 57 I do not understand the sentence; the virus was new, not unknown, and I am not clear what 'facilitated a more concrete response within	Changed 'unknown' to 'new' Changed 'facilitated a more concrete response within care homes' to 'facilitated a more

care homes' means.	objective and decisive response within care homes' to enhance clarity.
Page 18 line 41 what does the 'shifting roles and identities' mean?	Added 'a shift in roles and identities to take on NEWS related work'
Page 18 line 52 contact from a trusted NHS professional?	Yes – changed 'source' to 'professional'
Page 19 line 12 why would care home staff see an intervention that was introduced from outside their own sector as irrelevant or inappropriate?	It is completely possible that staff would think an intervention designed for use in hospitals would not be regarded as useful in care homes, as both the setting, and skill set of staff could be regarded as incompatible from a common sense perspective. This is not just a theoretical proposition, but can be seen in other studies of care home staff and is a view that is also shared across some of the wider professional community. We have included 'for use in care homes' to ensure this is clear, but we don't feel the statement requires any further explanation.
Page 19 line 31 I suggest revising the sentence about study not being representative. The study was conducted in a particular context and it is this that needs to be considered when looking at issues around transferability. A qualitative study would not be 'representative'.	Agreed. Revised 'representative' to 'reflect' to soften the language on this.
Page 19 line 40 could you clarify whether you set out to include GPs in your study. Did you try to recruit, but had no response?	Added 'despite our efforts' to clarify – we attempted to recruit GPs but had no response.
Page 19 line 55 what does 'offering reassurance and structure to their role' mean? – please clarify.	Added 'clinical' to clarify.

Reviewer: 2

Dr. LK Bragstad , Universitetet i Oslo Det medisinske fakultet

Comments to the Author:

Thanks for the opportunity to review the revised manuscript. I appreciate the authors' responses and changes made in response to the reviewer comments supplied from peer reviewers.

The manuscript has gained clarity, and is now much improved in my opinion. I appreciate the efforts made to describe the UK system for an international reader and particularly appreciate table 1 – this adds significant clarity for my reading. Well done on the revision.

I have only minor comments at this point, see comments below.

Comment raised	Response by author
Methods – Page 9, lines 13-15: The authors state that interviews were ceased when saturation was reached. Personally, I think saturation is a difficult concept, as I am not of the belief that additional interviews would not yield new data. However, my question is whether this was in fact the case – that you conducted transcriptions, and analysis while simultaneously conducting interviews (in May 2020). If that is the case, I think this would be an important piece of information to provide in the	Thank you, you are correct that we carried out interviews simultaneously (i.e. an iterative process between interviewing and analysis). We have added this information in the Methods section. We agree the term saturation is difficult and so we have also provided more detail regarding 'type' of saturation and provided and reference for this.

methods description. It would add to the transparency of the methods used.	
Results – At the end of the results section, as a reader I would appreciate a sentence or two to summarize rather than ending with two quotes. Maybe the first section of the discussion (summary of findings) should indeed be moved to the end of the results section?	It is not uncommon for sections to end with quotations, in fact some of the other sections higher up in the results section also end with quotations. However, we have moved a sentence to the end of the section before moving into the discussion to ‘round off’ this point. We did not feel it was appropriate to move the first section of the discussion, as this is an overall summary of the findings and traditionally this is placed at the start of the discussion.
Discussion/limitations Another thing I am curious about is whether the authors have given any thought to how the rapid roll-out of these tools and the guidance and supervision given to home care staff might also impact on the validity of the measurements. As you identify in our paper, these tools were originally developed to be used by a different group of professionals, with other competencies than the staff who are now using these. Any thoughts on this might be useful to share either in limitations or in discussion.	This is an interesting point, however unfortunately we don’t have data to support any concrete/data-based reflection on this, either in the results or discussion/limitations.
The lack for information from the care staff who used NEWS might also need to be highlighted as a possible limitation, even though you state that the perspectives you sought were from professionals at “higher” levels. My concern is that the administrative and leadership staff may have quite different experiences with these kinds of tools, this has been apparent in other studies evaluating similar tools. My general advice would be to highlight this as a limitation, rather than focus on the lack of information from the care home residents. Would you expect the care home residents of these homes to be able to discuss and give insight into your stated research questions?	We have added this as a limitation in the strengths and limitations section – we did seek to interview staff at all grades and ultimately interviewed three non-senior care home carers.
I notice that you have a statement about the inability to identify the active ingredients of the intervention. I find this statement a bit peculiar as long as this was not a goal stated in the purpose/aim. Furthermore, I would expect this to be a goal more connected to a different kind of evaluation, using other methods. Your informants wouldn’t, in my opinion, be the best source of information of this kind.	We have deleted this sentence to enhance clarity.
Page 18, lines 25-29: You refer to the inability to calculate the NEWS score when the measurements are not complete. I would have liked to see a bit more of your reflections around this. There are studies that indicate that the score itself is not the only important part these tools play. Using the tools	We feel that we have discussed NEWS scores in the context of individual measurements. However, we have included an extra few words, just to make that point clear in the text “NEWS score per se over and above the individual

to identify change in health status may also contribute valuable information for health care professionals using NEWS and similar EWS tools.	measurements”
--	---------------